# Seasonal Energy Flexibility Through Integration of Liquid Sorption Storage in Buildings

**Luca Baldini ***[ID] **and Benjamin Fumey**

Empa–Swiss Federal Laboratories for Materials Science and Technology, 8600 Dubendorf, Switzerland;
benjamin.fumey@empa.ch
**\*** Correspondence: luca.baldini@empa.ch

**Abstract:** The article estimates energy flexibility provided to the electricity grid by integration of long-term thermal energy storage in buildings. To this end, a liquid sorption storage combined with a compression heat pump is studied for a single-family home. This combination acts as a double-stage heat pump comprised of a thermal and an electrical stage. It lowers the temperature lift to be overcome by the electrical heat pump and thus increases its coefficient of performance. A simplified model is used to quantify seasonal energy flexibility by means of electric load shifting evaluated with a monthly resolution. Results are presented for unlimited and limited storage capacity leading to a total seasonal electric load shift of 631.8 kWh/a and 181.7 kWh/a, respectively. This shift, referred to as virtual battery effect, provided through long-term thermal energy storage is large compared to typical electric battery capacities installed in buildings. This highlights the significance of building-integrated long-term thermal energy storage for provision of energy flexibility to the electricity grid and hence for the integration of renewables in our energy system.

**Keywords:** long-term thermal energy storage; seasonal thermal energy storage; thermochemical energy storage; liquid sorption storage; power-to-heat; seasonal energy flexibility; seasonal load shifting; virtual battery effect

## 1. Introduction

There is an increasing need for energy storage in buildings to allow for integration of intermittent on-site renewable energy sources or the provision of energy flexibility to the electric grid. Thermal storage thereby plays an important role as it allows cheaply offsetting large amounts of energy. Thermal storage capital cost is low when compared to electric batteries and thus allows for economical energy storage also over longer time periods thus operation with substantially fewer cycles over its lifetime. At its extreme, seasonal storage is possible whereby the low number of cycles put a challenge on acceptable investment cost of the storage technology [1]. Seasonal storage is of great importance for significantly increasing renewable fraction in the operation of buildings. Without seasonal storage, excess energy available in summer cannot be made available for coverage of heat demand in winter such that renewable fraction, especially for space heating remains strongly limited. With seasonal energy storage, excess energy available from on-site renewable production or electricity from the grid can be absorbed in summer to be made available in winter when space heating demand is largest. Assuming a heat pump to provide space heating by using electricity, long-term storage thus contributes to significant load shifting from the winter into the summer. Power to heat conversion with the thermal energy storage together can be considered as a virtual electric battery and the electricity offset as the virtual battery effect. This consequently leads to an increase of integrated on-site renewables or provides a significant amount of energy flexibility to the electric grid. As the flexibility offered represents a seasonal rather than a short-term shift, it is referred to here as seasonal energy flexibility.

Currently, available sensible water storages or storages based on phase change materials suffer from a continuous heat loss such that seasonal storage is possible only at a large scale. To enable compact, long-term thermal energy storage on a building scale, higher volumetric energy storage densities are required, along with high in/out storage efficiencies, meaning low thermal losses during storage period. A class of energy storage known as thermochemical energy storage [2] promises to fulfill these criteria. Within this class there are different sub-classes often categorized into chemical reactions and sorption processes [3–6]. Depending on the type of reaction and the specific materials used energy densities between 200 and 2000 kWh/m$^3$ can be reached [6,7]. From measurements performed for a liquid sorption storage using aqueous sodium hydroxide (NaOH) as sorbent and water ($H_2O$) as refrigerant, a concentration difference from 50 to 27 wt.% NaOH was achieved for evaporation and absorber inlet temperatures of 5–20 °C and 25–38 °C, respectively, leading to a maximum theoretical volumetric energy density of 435 kWh/m$^3$ with reference to the diluted sorbent volume [8]. For comparison, volumetric energy storage density of water is at around 55 kWh/m$^3$, assuming a temperature difference of 50 K. Most important for thermochemical energy storage is the fact that energy is stored rather by means of a chemical potential than by actual thermal energy such that no continuous heat losses occur during the storage phase but only at conversion (charging/discharging). Generalized indication of storage density for thermochemical energy storage is problematic as it depends on many operational parameters, i.e. mainly temperatures. This fact is very often overlooked even in the research field of sorption energy storage. For a cross comparison of different thermochemical storage types, materials, and processes, thus a common reference framework as suggested, e.g., for the building application, in [9], is needed. A generalized metric provided in [10] can be used for performance comparison across different storage processes reported in literature but does not replace performance metrics such as volumetric energy density, asking for a uniform basis for evaluation.

There are several possibilities of integrating sorption storage into building energy systems, largely depending on the type of process, i.e. open or closed sorption process [5,11–13]. In an open process, absorbate for discharging is provided at atmospheric conditions, typically by ambient air and it is again released to the ambient in charging. In a closed process, phase change is typically taking place at subatmospheric conditions. The absorbate is provided through evaporation using a low temperature source in discharging and is again condensed using a low temperature sink in charging. The coupling between the storage and the heat source/sink happens through a heat exchanger. This article strictly focuses on a closed liquid sorption process using NaOH/$H_2O$ as sorption couple. An overview of examples of open and closed sorption storage processes is presented in [10].

Unlike a heat battery directly storing and releasing heat, sorption storage rather works as a heat pump, thus needing contact to two thermal reservoirs for charging or discharging. In charging a high temperature heat source is required for desorption (evaporation of water from the liquid sorbent, e.g., NaOH) as well as a low temperature heat sink for condensation of the water vapor extracted from the sorbent. In discharging, a low temperature heat source is needed for evaporation of water and a medium temperature sink, typically the building, for absorption of the water vapor. For integration of a sorption storage into the building energy system, it can be coupled with solar thermal collectors for the high temperature source as well as with a ground heat exchanger for low temperature heat source and sink [14]. More interesting from an energy flexibility perspective is the coupling of the storage with the electric grid through a compression heat pump. The heating system can then be looked at as a double-stage heat pump with one stage being a compression heat pump and another stage being a thermal or chemical heat pump, i.e. the sorption storage. Similar hybrid concepts combining compression and sorption cycles are presented in [15–17]. As an electricity source, on-site PV and/or the electric grid can be used. In charging mode, excess electricity can be received, while in discharging mode a smaller temperature lift is expected from the compression heat pump as it only represents one stage of the hybrid concept. Consequently, a higher heat pump efficiency and thus lower electricity consumption is expected. This article strictly focuses on the second option of double-stage heat



pumping in order to address the seasonal energy flexibility offered to the electricity grid through building integration of a liquid sorption storage.

## 2. Materials and Methods

### 2.1. Sample Building

For the assessment of the seasonal energy flexibility by using a liquid sorption storage, a sample single family home (SFH) with 140 m$^2$ of floor area is considered as defined by the IEA SHC Task 44 /HPP Annex 38 reference framework [18]. Out of three SFH buildings presented there, the SFH45 was picked, showing an area specific annual space heating demand of 46.255 kWh/(m$^2$·a). From three different climatic locations proposed, only the one for Strasbourg, France was considered. Results from detailed building simulations are provided as monthly values. This monthly resolution is sufficient to capture seasonal energy flexibility. From simulation results presented, monthly space heating loads, the instantaneous design heat load and average supply and return temperatures of the floor heating system were used (Table 1). Monthly average ambient temperatures for the location of Strasbourg were taken from [19] and are also shown in Table 1. Ambient temperature data was relevant for the evaluation of the air-source heat pump as well as for the sorption storage performance.

**Table 1.** Monthly average space heating load, space heating supply and return temperatures for SFH45 under climate of Strasbourg from [18] and ambient temperatures from [19].

| Input Variables | Jan. | Feb. | Mar. | Apr. | May | Jun. | Jul | Aug. | Sep. | Oct. | Nov. | Dec. |
|---|---|---|---|---|---|---|---|---|---|---|---|---|
| SH load (kWh/m$^2$) | 11.99 | 8.18 | 4.53 | 0.96 | 0.025 | 0 | 0 | 0 | 0 | 1.55 | 7.62 | 11.4 |
| Tsup (°C) | 30.1 | 29.1 | 27.4 | 26.1 | 27.2 | 0 | 0 | 0 | 0 | 26.4 | 28.1 | 29.4 |
| Tret (°C) | 22.2 | 21.4 | 20.7 | 20.4 | 20.5 | 0 | 0 | 0 | 0 | 20.5 | 21 | 22 |
| Tamb (°C) | 0.9 | 2.4 | 6.1 | 9.7 | 13.8 | 17.2 | 19.2 | 18.6 | 15.7 | 10.7 | 5.3 | 2.1 |

### 2.2. System Description

For the building integration of the closed, liquid sorption storage, a combination with a compression heat pump was chosen. In this way the sorption-based thermal energy storage is coupled to the electricity grid through the compression heat pump. The heat and mass exchanger (HMX) of the sorption storage is shown in Figure 1 with the absorber/desorber representing the left chamber and the evaporator/condenser the right chamber in the figure, respectively.

The heat pump is a water/water heat pump with an additional water/air heat exchanger. The schematic of storage integration is shown in Figure 2a in charging operation and Figure 2b in discharging operation. The H-shaped component represents the HMX. The sorption storage tanks consisting of absorbate (water), concentrated sorbent (NaOH) and a diluted sorbent tank, respectively, are not shown here. For more details of the sorption storage system with all components included, it is referred to [14,21].

In charging operation (Figure 2a), the compression heat pump is providing cold to the condenser of the HMX, while providing heat to the desorber of the HMX. In order to balance the mismatch between the evaporator and condenser, power provided by the heat pump the condenser side is additionally connected to the water/air heat exchanger for excess heat rejection to the ambient. Alternatively, excess heat can be used for domestic hot water production instead. In discharging (Figure 2b), the heat pump is used to extract heat from the ambient air using the water/air heat exchanger while providing heat to the evaporator of the HMX. In the absorption process taking place in the absorber of the HMX, heat is released to the building for space heating purposes.

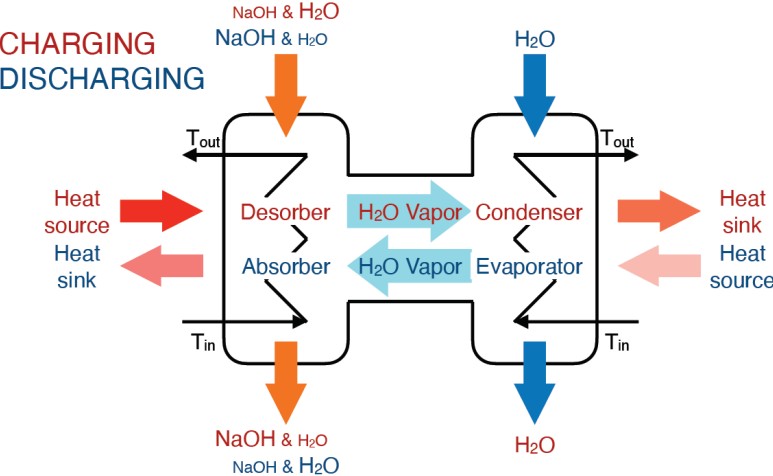

**Figure 1.** Heat and mass exchanger (HMX) with absorber/desorber on the left side and evaporator/condenser on the right side being connected to each other for water vapor exchange. During charging, diluted sorbent enters the desorber from the top leaving it as concentrated sorbent at the bottom. Thereby, water is evaporated by the external heat source and transported to the condenser, where it changes back to its liquid state, releasing heat to a respective sink. In charging, concentrated sorbent enters the absorber at the top, leaving it as diluted sorbent at the bottom. Thereby, water being evaporated by a low temperature heat source in the evaporator is absorbed, releasing useful heat for space heating or domestic water production. Heat exchange between sorbent and heat transfer fluid is followed in counterflow, allowing for the optimal exploitation of available temperatures. Adapted from [20].

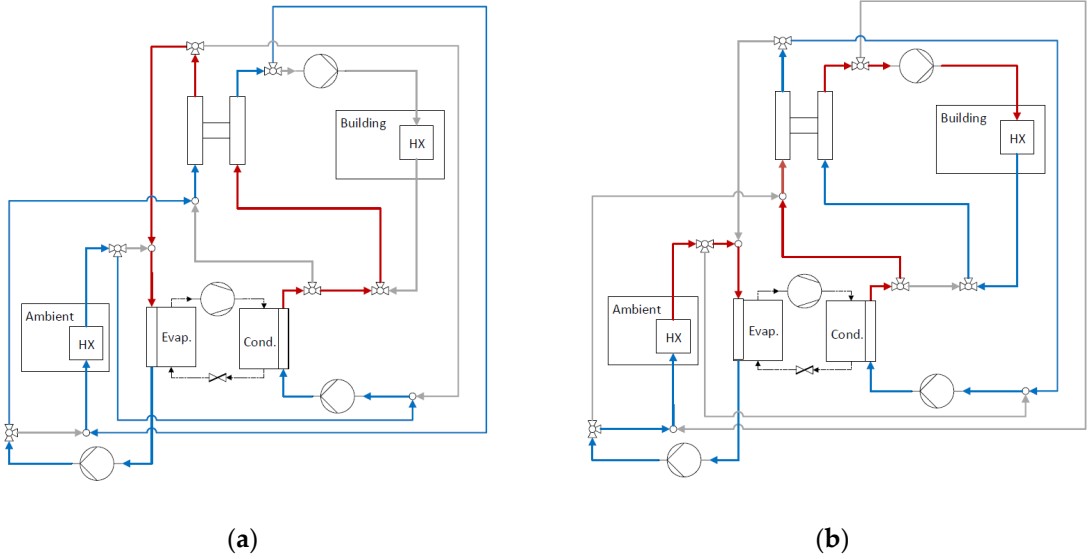

(**a**)                    (**b**)

**Figure 2.** Schematic of the sorption storage integration together with a compression heat pump, where the H-shaped component represents the HMX: (**a**) Charging mode: Heat pump provides high temperature for desorption and low temperature for condensation; (**b**) Discharging mode: Heat pump provides low temperature heat for evaporation.

### 2.3. Liquid Sorption Storage Modelling

The liquid sorption process is modeled, assuming thermodynamic equilibrium. This means that for a given absorbate vapor pressure and temperature of the liquid sorbent the equilibrium concentration is determined. In charging operations, a fixed outlet sorbent concentration of 50 wt.% NaOH and a maximum desorption temperature of 55 °C are chosen. For this state the water vapor pressure present

in the desorber, as well as in the condenser, is determined and with it the resulting condensation temperature. In discharging, the source temperature for the evaporation determines the absorbate pressure in the evaporator and absorber respectively and with it the sorbent concentration depending on its temperature. Minimum sorbent concentration in discharging is thus determined by the absorbate pressure given by the evaporator and the return temperature from the space heating system of the building entering the absorber. NaOH properties depending on temperature or concentration are calculated using available mathematical correlations [22].

For simplicity, during charging, heat provided to the desorber is assumed to be equal to heat rejected by the condenser, meaning that heat losses appearing in the desorber compensate for the neglected heat of solution.

## 2.4. Seasonal Flexibility Analysis

Seasonal flexibility offered by a building integrated sorption storage is evaluated based on its ability to use grid electricity in summer during charging and to reduce electricity demand in winter during discharging of the storage. Electricity is not stored seasonally but rather the potential to reduce electricity demand in winter/heating season. The electric load shift achieved is referred to as the virtual battery effect as it acts similarly to storing this part of electricity over the season. The seasonal energy flexibility must be separated in two parts, i.e., the additional electricity absorbed during charging (negative flexibility) and the electricity savings in discharging (positive energy flexibility). These energy flexibilities are quantified with reference to the regular heat pump operation without sorption storage integrated. The electricity consumed by the heat pump is calculated as the energy provided by the heat pump divided by its coefficient of performance (COP). The COP is calculated as the ideal Carnot COP based on thermal reservoir temperatures multiplied by a fixed isentropic efficiency ($\eta_{isen}$) of 0.5.

Negative energy flexibility provided with storage is equal to the electricity utilized by the heat pump to charge the storage, given a certain time ($\Delta t_{excess}$) with excess electricity available from the electricity grid (Equation (1)). During charging, the heat pump must provide the entire temperature lift between the HMX desorber and condenser and sensible heat stored in the charged, concentrated sorbent is lost ($\eta_{charging}$). The temperature of the desorber is fixed at 55 °C and the saturation pressure of water vapor at this temperature and a targeted sorbent concentration of 50 wt.% determines the condensation temperature needed in the HMX. The inlet to the condenser of the HMX provided by the evaporator of the heat pump then needs to be lower by the amount of the temperature difference between the absorbate and the heat transfer fluid ($\Delta T_{hx,HMX}$) and an additional temperature difference of the heat transfer fluid across the HMX condenser ($\Delta T_{HMX,ev}$). The evaporation temperature of the heat pump is then below the inlet temperature on the HMX condenser by $\Delta T_{hx,hp}$. In the condenser of the compression heat pump there is no temperature difference assumed between condensation temperature and temperature of the heat transfer fluid leaving the condenser. This assumption is justified because of the desuperhating taking place in the condenser, allowing even for secondary fluid temperatures above condensation temperature at the outlet of a counterflow-type condenser. This affects Equations (2) and (4) where condensation temperature of the heat pump is equal to the HMX desorber inlet temperature ($T_{desorption} + \Delta T_{hx,HMX}$) and $T_{supply}$ of the space heating, respectively.

$$E_{flex,charging} = P_{el,hp,charging}\Delta t_{excess} = \eta_{charging}\frac{Q_{hp,\,desorption}}{COP_{hp,charging}}\Delta t_{excess} \tag{1}$$

$$COP_{hp,charging} = \eta_{isen}\frac{T_{desorption} + \Delta T_{hx,HMX}}{\left(T_{desorption} + \Delta T_{hx,HMX}\right) - \left(T_{sat,p_{desorb}} - \Delta T_{hx,HMX} - \Delta T_{HMX,ev} - \Delta T_{hx,hp}\right)} \tag{2}$$

In discharging, the energy flexibility is calculated as the difference in electricity consumed during heating season ($\Delta t_{heating}$) by the single-stage and double-stage heat pump including storage respectively (Equation (3)). In case a sorption storage is included the temperature lift is to be provided by the compression heat pump is reduced, thus increasing its COP.

$$E_{flex,discharging} = (P_{el,ref,heating} - P_{el,storage,discharging})\Delta t_{heating} = Q_{heating}\left(\frac{1}{COP_{ref,heating}} - \frac{1}{COP_{storage,discharging}}\right)\Delta t_{heating} \quad (3)$$

In standard operation without storage installed, the heat pump COP ($COP_{ref,heating}$) is calculated based on supply temperatures ($T_{supply}$) of the space heating system and the evaporation temperature (Equation (4)). The latter is determined by the ambient air temperature reduced by the temperature difference between the air inlet of the water/air heat exchanger and the water outlet of the heat pump evaporator ($\Delta T_{air}$) and the temperature difference between the water and refrigerant ($\Delta T_{hx,hp}$) in the evaporator.

$$COP_{ref,heating} = \eta_{isen}\frac{T_{supply}}{T_{supply} - (T_{amb} - \Delta T_{air} - \Delta T_{hx,hp})} \quad (4)$$

When combined with the sorption storage the evaporation temperature of the heat pump remains the same but the condensation temperature of the heat pump being the inlet temperature of the HMX evaporator ($T_{HMX,ev,in}$) changes, leading to a different COP ($COP_{storage,discharging}$) as expressed in Equation (5). The inlet temperature to the evaporator side of the HMX provided by the compression heat pump as the first stage shown in Equation (6) is determined by the required supply temperature ($T_{supply}$) of the space heating and the available temperature lift from the sorption storage ($\Delta T_{50,max}$) between the maximum sorbent temperature and the evaporation temperature. Further, there is the temperature difference of the heat exchanger, once between the sorbent and heat transfer fluid ($\Delta T_{hx,HMX}$) and once between the absorbate and the heat transfer fluid ($\Delta T_{hx,HMX}$) and the temperature difference across the evaporator of the HMX ($\Delta T_{HMX,ev}$). The temperature lift provided by the sorption storage ($\Delta T_{50,max}$) depends on the maximum sorbent concentration of 50 wt.% as shown in Figure 3. In this analysis, a departure from ideal equilibrium condition is assumed, leading to a reduction of the effectively provided temperature lift.

$$COP_{storage,discharging} = \eta_{isen}\frac{T_{HMX,ev,in}}{T_{HMX,ev,in} - (T_{amb} - \Delta T_{air} - \Delta T_{hx,hp})} \quad (5)$$

with

$$T_{HMX,ev,in} = T_{supply} + \Delta T_{hx,HMX} - \Delta T_{50,max} + \Delta T_{hx,HMX} + \Delta T_{HMX,ev} \quad (6)$$

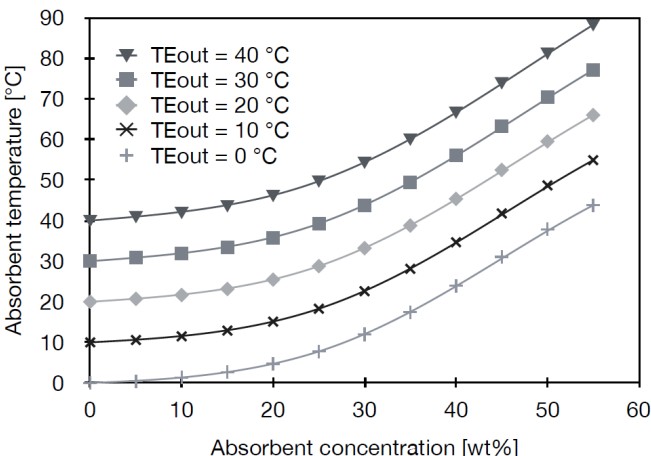

**Figure 3.** Ideal temperature lift provided between HMX evaporator temperature (TEout) and the maximum HMX absorber temperature by the sorption storage depending of maximum sorbent concentration. Adapted from [20].

The positive energy flexibility provided during discharging is depending on the amount of energy stored. This amount is determined during the charging phase. Maximum energy storage capacity is determined by the installed heat pump capacity multiplied by the time with available excess

electricity from the electricity grid ($\Delta t_{excess}$). In discharging mode, the fraction of space heating demand that can be covered by the stored energy is determined in order to identify the system operating in double-stage mode. When storage capacity is exhausted it switches to standard single-stage operation mode. This way the positive energy flexibility offered with limited storage capacity is calculated and confronted with the ideal case of unlimited storage capacity.

In order to determine the volumetric energy density of the sorption storage with reference to the diluted sorbent and the volume required to store the calculated energy, terminal sorbent concentrations were evaluated based on system temperatures. Specifically, for the HMX evaporator temperature, the saturation pressure of water vapor is calculated, determining the achieved terminal concentration in the HMX absorber based on absorber inlet temperatures. These temperatures are determined by the return temperatures from the space heating system plus a temperature difference ($\Delta T_{hx,HMX}$) between the heat transfer fluid and the sorbent. An average terminal sorbent concentration is then calculated over the heating period in order to calculate the sorbent mass needed to store the energy provided during charging. For this purpose an energy balance, a species mass (NaOH), and total mass balance over the HMX absorber is applied resulting in an expression for the inlet mass of concentrated sorbent depending on the known ratio of inlet to terminal concentration and the specific enthalpies of concentrated ($h_{sorbent,in}$) and diluted ($h_{sorbent,out}$) sorbent, respectively, and enthalpy of condensation for water ($h_{lg}$) given in Equation (7).

$$M_{sorbent,in} = \frac{Q_{heating}}{h_{sorbent,in} + \left(\frac{c_{sorbent,in}}{c_{sorbent,out}} - 1\right)h_{lg} - \left(\frac{c_{sorbent,in}}{c_{sorbent,out}}\right)h_{sorbent,out}} \tag{7}$$

The outlet mass of diluted sorbent then becomes

$$M_{sorbent,out} = \left(\frac{c_{sorbent,in}}{c_{sorbent,out}}\right)M_{sorbent,in} \tag{8}$$

The volumetric energy density of the sorption storage is calculated dividing the leaving sorbent mass ($M_{sorbent,out}$) by the sorbent density at the terminal concentration ($c_{sorbent,out}$) assuming a fixed temperature of 25 °C, representing the average absorber inlet temperature.

In Table 2, parameter values used for the evaluation of energy flexibility are summarized.

**Table 2.** Parameter values used for evaluation of energy flexibility provided with the building integrated sorption storage.

| Input Parameters | | | | | |
|---|---|---|---|---|---|
| $\eta_{isen}$ | 0.5 | $c_{sorbent,in}$ | 0.5 | $\Delta T_{hx,HMX}$ (K) | 3 |
| $\eta_{charging}$ | 0.9 | $T_{desorption}$ (°C) | 55 | $\Delta T_{hx,hp}$ (K) | 3 |
| $\Delta t_{excess}$ (h) | 720 | nominal heat pump/HMX capacity (kW) | 4.07 | $\Delta T_{HMX,ev}$ (K) | 3 |
| $\Delta T_{air}$ (K) | 10 | $\Delta T_{50,max}$ (K) | 25 | | |

## 3. Results

In the following, results are presented for the parameter settings according to Table 2. These include COP improvements, electricity demands, electric load shifting, i.e. energy flexibilities provided when using a building-integrated sorption storage. Results are presented for the different cases of (a) unlimited storage capacity and (b) limited storage capacity with reference to no storage installed. As an extension to the base case using input parameters according to Table 2 with 720 hours of available excess electricity, a simulation case assuming an increased number of 1080 hours of available excess electricity was considered as well.

*3.1. Coefficient of Performance (COP) Improvement and Electric Load Shifting, i.e., Seasonal Energy Flexibilities*

Sorption storage acts as a thermal heat pump, thus providing a certain temperature lift depending on maximum sorbent concentration. For a NaOH concentration of 50 wt.%, a temperature lift of around 38 K is ideally provided (Figure 3). This maximum theoretical temperature lift is determined by the difference in water vapor saturation temperature for pure water and sorbent solution respectively. Available temperature lift in reality is significantly lower due to imperfections of the process (deviations from equilibrium conditions) leading to an assumed temperature lift available of 25 K based on experience from experiments performed [8]. When using a sorption storage along with a heat pump, the temperature lift to be provided by the heat pump is reduced with comparison to heat pump operation without storage. According to Figure 4, heat pump COP is more than doubled when the total temperature lift required becomes small at moderate ambient air temperatures and is increased by about 50% during the coldest months in January and December.

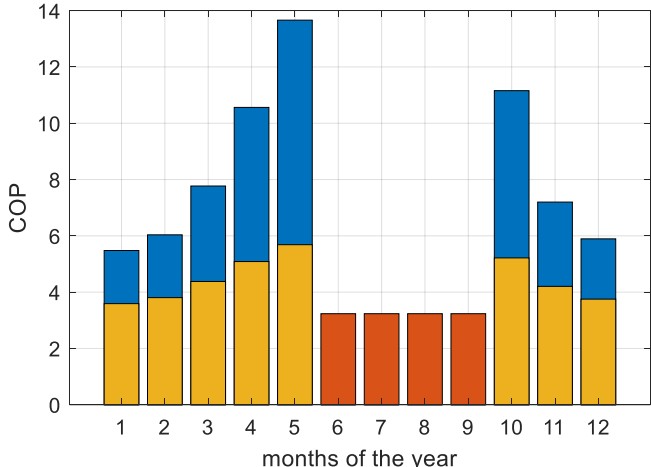

**Figure 4.** Coefficient of performance (COP) with (blue) and without (orange) sorption storage in heating operation and charging operation (red).

Improved COP with integrated sorption storage directly translates into electricity demand reduction in discharging representing positive energy flexibility to the grid. In Figure 5, electricity demands are shown for (a) no storage capacity limit and (b) limited storage capacity. The limitation in storage capacity is determined by the available heat pump capacity, the charging efficiency, and the total time of available excess electricity. In the proposed integration of the sorption storage with the heat pump, the latter provides the high temperature source and the low temperature sink at the same time when operating in charging mode. As a consequence, heat pump capacity available for charging is dictated by the evaporator power. The difference between condenser and evaporator power of the heat pump cannot be used by the sorption storage and is rejected over the water/air heat exchanger to the ambient air or used for domestic hot water production. With unlimited storage, capacity electricity demand is reduced during the entire heating operation (Figure 5a) while with the limited storage capacity, a demand reduction can be achieved only in January and a little bit in February (Figure 5b). When more excess electricity from the grid is available, charging duration and thus storage capacity can be increased such that more significant reduction in electricity demand can also be achieved in February (Figure 5c). This increase in available excess electricity does of course not have any effect on the theoretical case of unlimited storage capacity, which assumes full coverage of space heat demand by the sorption storage.

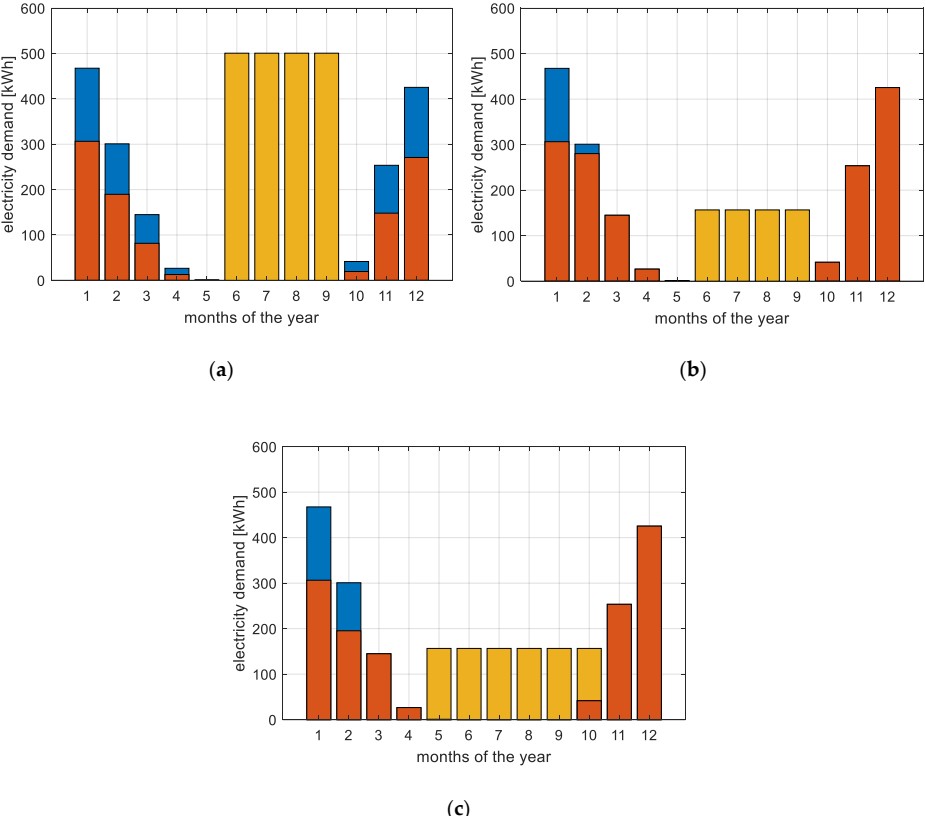

**Figure 5.** Monthly electricity demand of heat pump for space heating operation (discharging) with (red) and without (blue) sorption storage and charging of sorption storage (orange). For base case with 720 hours of excess electricity: (**a**) with unlimited storage capacity; (**b**) with limited storage capacity and alternative case with 1080 hours of excess electricity available; (**c**) with limited storage capacity.

Electric load shifted between seasons are shown in Figure 6. With unlimited storage capacity total seasonal load shift/flexibility amounts to 631.8 kWh/a, while with a limited storage capacity it reduces to 181.7 kWh/a for the base case with 720 hours of available excess electricity. This leads to 38% and 10.9% of electric load shifted for unlimited and limited storage capacity, respectively. When assuming 1080 hours of available excess electricity, the seasonal load shift increases to 266.7 kWh/a.

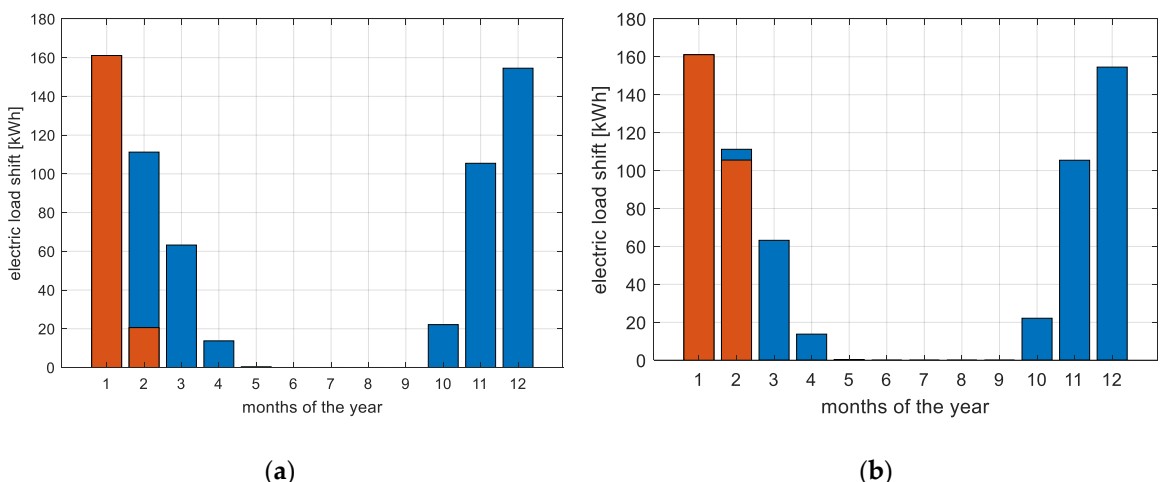

**Figure 6.** Electric load shift of heat pump operation with storage. With limited (red) and unlimited (blue) storage capacity for: (**a**) base case with 720 h of excess electricity and (**b**) case with 1080 h of excess electricity available.

## 3.2. Storage Capacity and Size

Based on temperatures assumed and resulting sorbent concentration lift, a volumetric energy storage density of 312.5 kWh/m$^3$ is achieved with reference to the diluted sorbent. With available charging capacity of the heat pump and assumed operation hours, a total energy storage capacity of 1822.6 kWh results, leading to a storage volume of the diluted sorbent of 5.8 m$^3$.

## 4. Discussion

The analysis performed captures seasonal load shifting/flexibility for a single-family home with an integrated sorption storage. It thus represents the electricity saved in heating operation because of seasonal thermal energy storage, called virtual battery effect. It acts the same as using an electric battery to store electricity seasonally. The latter is not done as it is economically not viable to use electric batteries for seasonal storage. This is hence a valuable feature of thermal energy storage. The virtual battery capacity achieved with the limited storage capacity is equal to 181.7 kWh. When comparing this with a standard capacity of an electric battery installed in single family homes of around 8 kWh, the sorption storage represents a 22.7 fold battery capacity. When comparing to the unlimited sorption storage capacity, a 79 fold battery capacity results. The limited sorption storage capacity leads to a storage volume of the diluted sorbent of 5.8 m$^3$. When no dead volume is assumed this represents the full storage volume needed. When using fixed storage tanks for concentrated and diluted sorbent as well as for the absorbate a total storage volume of 11.6 m$^3$ would be required. In either case, this size could be fitted in an SFH, taking up a floor area of 2.9 and 5.8 m$^2$, respectively, assuming cubic tanks with a height of 2 m.

As shown in Figure 3, COP of the electric heat pump can be improved substantially when combined with a sorption storage. In the model a fixed isentropic efficiency was used, leading to some overestimation at very low temperature differences between evaporation and condensation. In reality, isentropic efficiency is not constant but varies across the operation range of a heat pump. When designed accordingly, however, high and relatively constant isentropic efficiencies can be achieved at low temperature lifts for a limited operation window [23].

COP improvement is the main driver for electric load shifting and reflects the effect of the double-stage heat pumping when introducing a sorption storage along with an electric heat pump. The other important part is the available energy storage capacity. It was differentiated between limited and unlimited capacity. Energy flexibility offered with limited storage capacity is smaller than expected because of limitations present in charging. When sizing a heat pump for a low-energy house such as the SFH45, installed capacity is rather small. In a hybrid operation of heat pump and sorption storage, as suggested, all heat provided to the building or the storage needs to go through the heat pump, making it to become the bottleneck. Another restriction comes from the time with available excess electricity from the grid. This is a given and strongly depends on the specific electricity generation system considered.

Different possibilities are seen to increase energy flexibility. Heat pump and HMX capacity could be increased, leading to overcapacity installed. When using a capacity-controlled heat pump, this would not negatively impact space heating operation but only system cost. Alternatively, only HMX size could be increased, dimensioning it rather for the charging than for the discharging operation being determined by the space heating demand of the building. In order to make use of larger HMX capacity, an additional heat source such as solar thermal collectors would be needed. While this would provide additional heat for charging it would not increase the negative energy flexibility. As it leads to larger energy storage capacity and hence seasonal load shift, it would still increase positive energy flexibility offered during heating season. Further, extended time of available excess electricity from the grid could be assumed, leading to larger energy storage capacity. The latter could be justified as times with excess electricity will increase with an increasing share of renewables in the electricity grid. Currently, excess is assumed for the months of June to September with an average of 6 h a day. When considering energy production profiles of solar thermal collectors or PV installations,

excess production with reference to space heating demand could already appear in April and May as well as in October. When assuming, for example, 6 instead of 4 months with excess electricity, raising the number of hours by 50%, storage capacity and consequent seasonal electric load shift would rise to 266.7 kWh/a or 16.1% with reference to no storage. This increase of time with excess electricity assumed can be seen as a proxy for any measures discussed above to increase charging and thus energy storage capacity.

Looking at the overall storage potential of building integrated sorption storage for the case of Switzerland, assuming that an average storage capacity of 266.7 kWh is installed in every one of the 1.7 millions of domestic buildings (including multi-family homes), a total figure of 0.46 TWh would result, representing 5.04% of existing electric storage capacity in Switzerland provided by seasonal hydro storage [24].

Building-level sorption storage has substantial potential to provide energy flexibility to the electricity grid and support the integration of renewables. For a better prediction of available energy flexibility a more detailed analysis based on an improved model of the building and the sorption storage with higher temporal resolution is needed. Further, an extended analysis of design parameters and their influence on storage performance together with an optimized operation for minimal electricity demand and $CO_2$ emissions are desirable. These additions to the current study will be addressed in future research.

**Author Contributions:** Conceptualization, L.B.; Formal analysis, L.B.; Investigation, L.B. and B.F.; Methodology, L.B.; Writing – original draft, L.B.; Writing – review & editing, L.B. and B.F. All authors have read and agreed to the published version of the manuscript.

**Funding:** This research work was financially supported by the Swiss Innovation Agency Innosuisse grant Nr. 1155002545 and is part of the Swiss Competence Centre for Energy Research SCCER HaE. Supplementary funding was received from the Swiss Federal Office of Energy SFOE grant Nr. SI/501605-01 in the frame of the IEA SHC Task 58/ECES Annex 33 participation.

**Conflicts of Interest:** The authors declare no conflict of interest. The funders had no role in the design of the study; in the collection, analyses, or interpretation of data; in the writing of the manuscript, or in the decision to publish the results.

## Nomenclature

| | |
|---|---|
| $E_{flex,charging}$ | Negative energy flexibility offered during charging (kWh) |
| $E_{flex,discharging}$ | Positive energy flexibility offered during discharging (kWh) |
| $P_{el,hp,charging}$ | Electric power of heat pump during charging (kW) |
| $P_{el,storage,discharging}$ | Electric power of heat pump during discharging (kW) |
| $P_{el,ref,heating}$ | Electric power of heat pump without storage during heating (kW) |
| $\Delta t_{excess}$ | Time of excess electricity available (hours) |
| $\Delta t_{heating}$ | Time with space heating demand (hours) |
| $\eta_{charging}$ | Charging efficiency (%) |
| $\eta_{isen}$ | Isentropic efficiency of heat pump (%) |
| $Q_{hp,desorption}$ | Heat provided by heat pump during charging of storage (kWh) |
| $Q_{heating}$ | Space heating demand (kWh) |
| $COP_{hp,charging}$ | Heat pump COP during charging |
| $COP_{storage,discharging}$ | Heat pump COP during discharging |
| $COP_{ref,heating}$ | Heat pump COP without storage during heating |
| $T_{desorption}$ | Sorbent temperature in HMX desorber during charging (°C) |
| $T_{sat,p\_desorb}$ | Saturation temperature of water at desorption pressure (°C) |
| $T_{supply}$ | Supply temperature of space heating system (°C) |
| $T_{amb}$ | Ambient air temperature (°C) |
| $T_{HMX,ev,in}$ | Inlet temperature at HMX evaporator (°C) |
| $\Delta T_{hx,HMX}$ | Temp. difference primary/secondary fluid in HMX heat exchanger (K) |
| $\Delta T_{hx,hp}$ | Temperature difference primary/secondary fluid in hp heat exchanger (K) |
| $\Delta T_{HMX,ev}$ | Temperature difference of HTF across HMX evaporator/condenser (K) |

| | |
|---|---|
| $\Delta T_{air}$ | Temperature difference ambient air inlet to evaporator water outlet (K) |
| $\Delta T_{50,max}$ | Maximum temperature lift sorption storage (K) |
| $M_{sorbent,in}$ | Concentrated sorbent mass entering HMX absorber in discharge (kg) |
| $M_{sorbent,out}$ | Diluted sorbent mass leaving HMX absorber in discharge (kg) |
| $h_{sorbent,in}$ | Specific enthalpy of sorbent entering HMX absorber (kJ/(kg·K)) |
| $h_{sorbent,out}$ | Specific enthalpy of sorbent leaving HMX absorber (kJ/(kg·K)) |
| $h_{lg}$ | Specific enthalpy of condensation for water (kJ/(kg·K)) |
| $c_{sorbent,in}$ | NaOH concentration of sorbent entering HMX absorber (kg/kg) |
| $c_{sorbent,out}$ | NaOH concentration of sorbent leaving HMX absorber (kg/kg) |

## Abbreviations

| | |
|---|---|
| COP | Coefficient of performance |
| HMX | Heat and mass exchanger |
| hp | Heat pump |
| $H_2O$ | Water |
| NaOH | Sodium hydroxide |
| PV | Photovoltaics |
| SFH | Single family home |
| SFH45 | Single family home with specific space heating demand of 45 kWh/m$^2$ |

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
