# Peer review of "Seasonal Energy Flexibility Through Integration of Liquid Sorption Storage in Buildings"

_energies, doi:10.3390/en13112944_

Round 1
Reviewer 1 Report
This article presents a study of a combined liquid sorption storage with a compression heat pump as a source for energy storage for a house in Strasbourg, France, over an entire year. I read similar studies for houses in Italy, published in the same journal last year. But there was the accent more on the simulation than on monitoring the efficiency as a function of temperature difference. It is very interesting that also the changes due to the season (winter or summer) are included, and the results are compared to a typical battery that normally installed in some buildings. I find this study very interesting and useful for those that are dealing to find optimal and improve renewable energy storage systems. Therefore, I think this paper can be accepted for publication in its present form.
Author Response
The reviewer provides a concise summary of this article and correctly points to its major goals and contributions.
We thank the reviewer for his positive feedback and are glad to hear that he sees added value by the presented study.
Reviewer 2 Report
Article:
Seasonal energy flexibility through integration of liquid sorption storage in buildings by Baldini and Fumey.
Summary:
The article estimates energy flexibility provided to the electricity grid by integration of long10 term thermal energy storage in buildings. To this end, a liquid sorption storage combined with a 11 compression heat pump is studied for a single family home.
The article is according to the referee of a high quality and very well structured. There are a couple of minor concerns which can be addressed in the revised manuscript before publication.
Comments:
- Lines 52 to 56: Measurements are briefly discussed of the volumetric energy density of an aqueous sodium hydroxide system coming to a value of 435 kWh/m3, which is indeed very large compared to sensible heat storage in water with a temperature lift of 50K. However, when presenting volumetric energy density values one should always precisely indicate which volume is taken as the reference. In an honest comparison one should calculate the volumetric values based on the system volume.
- Line 116: Figure 1 is a very nice sketch of the heat and mass exchanger but it would be nice to extend the caption somewhat with some more explanation.
- Line 132: Please indicate also in the caption that the H-shaped component represents the HMX.
- Line 138: It is stated that “In charging operation, a fixed sorbent concentration of 50wt% NaOH and a desorption temperature of 55°C are chosen”. Is the 50wt% some kind of mean value along the heat and mass exchanger?
- Line 247-248: It is not clear to the referee how to authors conclude based on Figure 3 that for an NaOH concentration of 50wt% a temperature lift of around 38 K is provided ideally.
- Lines 260-270: Discussion on the limitation in storage capacity is not clear to the referee. Why does it only play a role in January and February for the limited storage capacity case?
- Line 285: It is stated here that a storage volume of the diluted sorbent is needed of 5.8 m3. Is it not more fair to consider the whole system volume instead of only the volume for the diluted phase?
- Line 348-351: Is this part of the text on Author Contributions dictated by the editor?
Author Response
We thank the reviewer for the valuable feedback and hints to further improvements. We addressed his comments in the following way:
- We completely agree with your comment and now added: "storage density with reference to the diluted sorbent volume" on line 56
- Line 121, caption of Figure 1 extended, now also explaining the working principle:
"HMX with absorber/desorber on the left side and evaporator/condenser on the right side being connected to each other for water vapor exchange. During charging, diluted sorbent enters the desorber from the top leaving it as concentrated sorbent at the bottom. Thereby, water is evaporated by the external heat source and transported to the condenser, where it changes back to its liquid state, releasing heat to a respective sink. In charging, concentrated sorbent enters the absorber at the top leaving it as diluted sorbent at the bottom. Thereby, water being evaporated by a low temperature heat source in the evaporator is absorbed, releasing useful heat for space heating or domestic water production. Heat exchange between sorbent and heat transfer fluid is followed in counterflow, allowing for optimal exploitation of available temperatures. Adapted from [20]. - Ok, mentioning of H-shaped component being the HMX has now been added to the caption of Figure 2, line 145: "Schematic of the sorption storage integration together with a compression heat pump, where the H-shaped component represents the HMX: (a) Charging mode: Heat pump provides high temperature for desorption and low temperature for condensation; (b) Discharging mode: Heat pump provides low temperature heat for evaporation."
- On line 152 it is now referred to the 50wt% as outlet concentration and the 55°C as maximum desorption temperature to emphasize that this is a maximum achieved rather than an average. In caption of Figure 1 it is further pointed to the counterflow heat exchange in the HMX leading to maximum concentrations reached at the bottom of the desorber during desorption where heat transfer fluid enters with its maximum temperature.
- Thanks for the comment. On line 262 we now added further explanation on equilibrium conditions and maximum temperature lifts achieved for the sorbent used being NaOH by stating: "This maximum theoretical temperature lift is determined by the difference in water vapor saturation temperature for pure water and sorbent solution respectively." Also we added a reference to measurements performed on line 266, demonstrating the realistically achieved temperature lift being substantially below the theoretical one.
- With limited storage capacity available contributions from the energy storage to reduce electric loads in winter are restricted to January and February. If the storage capacity was larger, further contribution could be provided also in March and further months with space heating demand.
- We totally agree with you. As the study presented here is of theoretical nature though no reliable number of the effective system volume can be given. For this reason the diluted sorbent is used as reference and it is declared accordingly.
- This is as requested by the publisher and is generated based on author inputs when uploading the manuscript.
Reviewer 3 Report
The article “Seasonal energy flexibility through integration of liquid sorption storage in buildings” by Luca Baldini and Benjamin Fumey discusses the specific case of a single family home, equipped with a liquid sorption storage combined with a compression heat pump, in the context of the energy flexibility provided to the electricity grid by integration of longterm thermal energy storage in houses.
1 The article focuses on a specific case. It would be useful for the reader to include more specific data that has been used in this study. For example, ref 14 gives some numbers as approximates and also discusses that some of the numbers can be reduced. To enable the reader to use the model for their own purposes, more details are necessary. Please give specific sizes for the HXM system, maybe include them as supplementary data.
2 Please note that Figure 2 in ‘Benjamin Fumey et al. / Energy Procedia 46 (2014) 134 – 141’ [reference 14] is plotting the same data as Figure 3 in this manuscript, although the physical presentation of the graph has been changed. Please include a reference as these are not new results.
3 It would also be helpful for the reader to provide specific seasonal data that went into the model rather than only providing a reference that includes other data as well.
4 I could not access the link in [19], please check.
5 In line [250], please give a reference to the experimental data the manuscript is referring to.
6 In the discussion, the authors assert that:
“When assuming hence 6 instead of 4 months with excess electricity, raising the number of hours by 50%, storage capacity and consequent seasonal electric load shift would rise to 266.7 kWh/a or 16.1% with reference to no storage.”
Has this simulation been run as well? This should be part of the results. Please check the manuscript for consistency that the discussion only includes results that have been included in the results section with enough specific data for the reader to be able to re-do the calculations.
Overall, the manuscript needs revisions before being published. Please include enough information for the reader to follow the simulation.
Author Response
We thank the reviewer for his valuable feedback that we tried to address as following:
- More specific data is requested but it is unclear what it is referred to exactly. It is pointed to reference [14] and to numbers provided there but again, it is unclear which numbers are meant and what value they would add to the study. Due to lack of understanding, this point is not directly addressed. Maybe there is some misunderstanding that needs clarification: This paper presents a theoretical study using a simple mathematical model to assess the potential of combining a liquid sorption storage together with an air-source heat pump. Due to this theoretical nature, there are no practical considerations about the exact system design and individual component size besides the required storage volume, evaluated on a material/sorbent basis.
- This is done to avoid any copyright problems to occur but a reference to the original work is now provided. Addition to figure caption, Figure 3, line 221: "Adapted from [20]."
- The IEA SHC task document presents inputs data used to represent monthly space heating loads and supply and return temperatures. For convenience, the relevant inputs extracted are now explicitly listed in a new table (Table 1) on line 112, being introduced in the text on line 108.
- It was recognized that the web page with the provided link sometimes takes very long to load, making access difficult. Chosen values for the average monthly temperatures are therefore now also listed in Table 1 on line 112.
- Reference to paper [8] providing details on absorption/desorption experiments performed is now included on line 266.
- You are right, we are referring to a simulation that was not shown under results so far. A second set of figures for the case of 6 months (1080 hours) with excess electricity is now included in the results section such that it can be referred to in the discussion (newly added Figure 5c and Figure 6b) as requested. At this occasion we also adjusted the scales in the figures to facilitate comparison of results. Further, on line 256, we introduced a new case featuring an extended number of hours of available excess electricity by stating: "As an extension to the base case using input parameters according to Table 2 with 720 hours of available excess electricity, a simulation case assuming an increased number of 1080 hours of available excess electricity was considered as well."
On line 285 we comment on the effect of increasing available excess electricity by stating: "When more excess electricity from the grid is available, charging duration and thus storage capacity can be increased such that more significant reduction in electricity demand can also be achieved in February (Figure 5c). This increase in available excess electricity does of course not have any effect on the theoretical case of unlimited storage capacity which assumes full coverage of space heat demand by the sorption storage."
With the addition of missing model inputs in Table 1 it now should be possible for others to reproduce the study by using a similar mathematical model. Modelling of the sorbent side is based on equilibrium assumptions as described in section 2.3 and mass and energy balances applied as laid out in Equations 7 and 8. Details regarding the analysis of energy flexibility is provided in equations 1 to 6 such that to our understanding model reproducibility is now given.
Round 2
Reviewer 3 Report
My concerns have been addressed.
I would like to ask the authors to use a gender neutral pronoun when addressing reviewers, either referring to 'his/her comments' or to 'their comments'.